# Phytochemical Profiling and Antimicrobial Properties of Various Sweet Potato (*Ipomoea batatas* L.) Leaves Assessed by RP-HPLC-DAD

**DOI:** 10.3390/foods13172787

**Published:** 2024-09-01

**Authors:** Tasbida Sultana, Shahidul Islam, Muhammad Abul Kalam Azad, Md Jahurul Haque Akanda, Atikur Rahman, Md Sahidur Rahman

**Affiliations:** 1Department of Agriculture/Agricultural Regulations, University of Arkansas at Pine Bluff, 1200 North University Dr., 148 Woodard Hall, Mail Slot 4913, Pine Bluff, AR 71601, USArahmanm7226@uapb.edu (M.S.R.); 2Department of Physics and Astronomy, University of Arkansas at Little Rock, 2801 S University Ave., Little Rock, AR 72204, USA; arahman@ualr.edu

**Keywords:** *Ipomoea batatas* extracts, TPC, TFC, antioxidant, antimicrobial, HPLC-DAD

## Abstract

This study aimed to investigate the leaves of six cultivars of *Ipomoea batatas* L. from the USA, focusing on their Total Polyphenol Content (TPC), Total Flavonoid Content (TFC), antioxidant, and antimicrobial activities. TPC and TFC ranged from 7.29 ± 0.62 to 10.49 ± 1.04 mg TAE/g Dw, and from 2.30 ± 0.04 to 4.26 ± 0.23 mg QE/g Dw, respectively, with the highest values found in the ‘O’Henry’ variety. RP-High-Performance Liquid Chromatography identified six phenolic and flavonoid compounds: caffeic acid, chlorogenic acid, 3,5-dicaffeoylquinic acid, 3,4-dicaffeoylquinic acid, and quercetin, excluding gallic acid. The highest levels of these compounds were found in acidified methanolic extracts. Antioxidant activities, measured by ABTS and DPPH assays, showed low IC_50_ values ranging from 94.6 ± 2.76 to 115.17 ± 7.65 µg/mL, and from 88.83 ± 1.94 to 147.6 ± 1.22 µg/mL. Ferric Ion-Reducing Antioxidant Potential (FRAP) measurements indicated significant antioxidant levels, varying from 1.98 ± 0.14 to 2.83 ± 0.07, with the ‘O’Henry’ variety exhibiting the highest levels. The antimicrobial activity test included five Gram-positive bacteria, three Gram-negative bacteria, and two pathogenic fungi. *S. aureus*, *S. mutans*, *L. monocytogenes*, *E. coli*, *S. dysenteriae*, and *C. albicans* were most susceptible to the methanolic extract. This study underscores the impressive antioxidant and antimicrobial activities of sweet potato leaves, often discarded, making them a valuable source of natural antioxidants, antimicrobials, and other health-promoting compounds.

## 1. Introduction

The sweet potato (*Ipomoea batatas*), a perennial crop propagated vegetatively, belongs to the *Convolvulaceae* or Morning Glory family, encompassing over 600 species worldwide. One of the *Ipomoea* genera known worldwide is the yellow sweet potato (*Ipomoea batatas* (L.) *Lam*.). Although its common name suggests a kinship with potatoes, sweet potatoes are scientifically distinct from potatoes. They are roots, not tubers, and originated in Latin America over 5000 years ago [1]. The spread of sweet potatoes across the globe accelerated after Columbus arrived in the Caribbean in 1492. The Spanish introduced the crop to Europe, from where it spread to Africa, India, China, Indonesia, and southern Asia, likely aided by Portuguese traders. Between 1597 and 1609, sweet potatoes were introduced to Japan from China. They became widely cultivated as a famine-relief crop in various parts of Japan throughout the 17th and 18th centuries due to their high tolerance to extreme weather and cultivation conditions characterized by drought and heat stress [2].

Sweet potatoes are cultivated in over 100 countries, primarily in tropical regions, and are more extensively grown in developing countries than any other root crop. They are the seventh most abundant food crop globally [3]. The crop boasts around 7000 cultivars, with about 75% of its production in Asian countries. The plant proliferates rapidly, allowing for the harvesting of its leaves up to six times a year, contributing to a higher annual yield than other leafy vegetables [4]. All parts of the plant, including leaves, roots, and vines are edible and come in various types characterized by a spectrum of skin and flesh colors ranging from white to cream, yellow, orange, and purple.

Antioxidants are crucial in mitigating oxidative stress-related illnesses by enhancing the body’s immune defenses against infectious diseases and maintaining balance of life processes. Numerous studies have indicated that degenerative conditions like atherosclerosis, cancer, asthma, diabetes, neurodegenerative diseases, and osteoporosis are often linked to harmful reactions involving free radicals. Plants are a rich source of antioxidant compounds, mainly polyphenols, which effectively neutralize free radicals and protect cells from reactive oxygen species damage by chelating metals donating hydrogen atoms, a role involved in regulating signal transmission and enzyme function, and interrupting free radical chain reactions [5]. Polyphenols are a category of secondary metabolite, naturally synthesized by various plants and fungi. They are typically divided into four groups: phenolic acids, flavonoids, lignans, and stilbenes [6].

However, phenolic acids comprise two primary groups: hydroxycinnamic acids and hydroxybenzoic acids. Both are hydroxy derivatives of aromatic carboxylic acids, such as cinnamic and benzoic acids, responsible for organoleptic properties like sourness, bitterness, odor, and preservative characteristics. They differ according to the number and position of hydroxylation and methoxylation of the aromatic ring. The most common hydroxycinnamic acid is Caffeic Acid (CA). The most well-known bound hydroxycinnamic acids are Chlorogenic Acid (ChAs), a combination of CA and quinic acids [7]. ChAs are water-soluble compounds named by Clifford in 1985. They can be divided into Caffeoylquinic Acids (CQAs) with three isomers (3-, 4-, and 5-CQA), diCaffeoylquinic Acids (diCQAs) with three isomers (3,4-diCQA; 3,5-diCQA; 4,5-diCQA), feruloyl quinic acids, and p-coumaroylquinic acids [8,9]. Gallic acid is one type of hydroxybenzoic acid widely found in plants. Flavonoids are the most wide-ranging metabolites, including flavonols, flavones, flavanones, isoflavones, and anthocyanins [10]. The most common flavonoids are flavonols such as quercetin.

Sweet Potato Leaves (SPL) have been shown to enhance physiological defense mechanisms and antioxidant status. Regular consumption of SPL can protect against oxidative stress by preventing LDL lipid oxidation and DNA damage in lymphocytes, as well as influencing erythrocyte glutathione and plasma tocopherol levels. Consuming 200 g of cooked SPL for seven days increased plasma polyphenol levels, improved the ferric-reducing ability of plasma, and boosted immune functions by enhancing the lytic activity of NK cells and cytokine IL-2, enhancing IL-4 secretion, increasing cellular multiplication response of phagocytic cells, and reducing inflammatory markers like IL-6 [11].

Green SPL is particularly rich in quercetin, shown to have three times the antioxidant capacity of kaempferol. Fifteen anthocyanins have been identified in SPL, contributing to their antioxidant, antimutagenic, and anti-cancer properties. These compounds have demonstrated effectiveness in inhibiting the proliferation of several cancer cell types, including HCT-116 colorectal cancer, MCF-7 breast cancer, and lung cancer cells. Methanol extracts of SPL have also been shown to suppress the growth of prostatic cancer cells through mechanisms involving cell cycle regulation, apoptosis activation, and reduced clonogenic survival [12].

SPL have gained significant attention for managing various infectious diseases, underscoring their role in promoting human health. In Brazil, the leaves are used to treat infectious diseases and are recognized for their antidiabetic effects [13]. Furthermore, research has shown that sweet potato leaves possess properties that can suppress HIV replication [14,15]. Research on several other cultivars worldwide has shown sweet potato leaves’ antioxidant and nutraceutical benefits. Therefore, gathering detailed information about various cultivars is crucial to validate their health benefits. The objective of this study is to develop and validate methods for analyzing CA, ChA, 3,4-diCQA, 3,5-diCQA, quercetin, and gallic acid using RP-HPLC-DAD with two different extraction methods. Additionally, the study aims to quantify antioxidant activity through ABTS, DPPH, and FRAP assays, estimate TPC and TFC via UV spectrophotometry, and assess antibacterial properties against human pathogenic microorganisms in six commercially grown varieties of sweet potato leaves in the USA.

## 2. Materials and Methods

### 2.1. Plant Materials and Growth Conditions

This study analyzed leaf samples from six distinct sweet potato cultivars. These cultivars included ‘O’Henry’, ‘Covington’, ‘Vardaman’, ‘Beauregard’, ‘Georgia Jet’, and ‘Centennial’. The cultivation process began by planting slips (cuttings) at the University of Arkansas’s Pine Bluff’s agricultural field (34°15′05.0″ N, 92°01′36.0″ W, at an altitude of 68 m above sea level) in May 2023. After a 90-day growth period, the sweet potato leaves were harvested in September 2023, as illustrated in Figure 1. During the growing period from May to September, the mean air temperature ranged from 92 to 72 °F. The total precipitation during this period was 2.3 inches. For each cultivar, triplicate leaf samples were collected directly from the field at intervals of two weeks.

### 2.2. Materials

All standard compounds and solvents for the analytical studies were obtained from Fisher Scientific, Waltham, MA, USA. The microbial strains used included *Staphylococcus aureus* (ATCC BAA-1708™), *Staphylococcus epidermidis* (ATCC^®^ 35984™), *Streptococcus mutans* (ATCC^®^ 35668™), *Listeria monocytogenes* (ATCC^®^ 19115™), *Bacillus subtilis* (Culti-Loops™ ATCC™ 6633™), *Escherichia coli* (ATCC^®^ 8793™), *Shigella dysenteriae* (ATCC^®^ 13313™), *Klebsiella aerogenes* (NCIMB 10102), *Candida albicans* (ATCC^®^ 14053™), and *Aspergillus niger* (ATCC^®^ 16888™). These were sourced from Microbiologies (St. Cloud, MN, USA) and Thermo Scientific (Lenexa, KS, USA). All chemicals were of analytical grade, and Milli-Q Direct-Q^®^ 3 UV Water Purification System (Darmstadt, Germany) purified water was used in all experiments.

### 2.3. Moisture Content Analysis

Upon collection, the leaves were immediately rinsed with distilled water to remove any surface contaminants. After air drying, the fresh leaves were weighed on a digital balance (A&D Engineering, Inc., Ann Arbor, MI, USA), and noted as fresh weight. The samples were then frozen at −80 °C for 24 h. Following the initial freezing, the samples underwent a 48-h freeze-drying process using a MillRock Technology Freeze Dryer (Model MD3053, Kingston, NY, USA) to eliminate residual moisture. The dry leaf samples were weighed and recorded as dry weight. The leaf moisture contents were calculated as follows [16]: $$\mathrm{Moisture}\;\mathrm{content}\;(\%)=\frac{\mathrm{Fresh}\;\mathrm{weight}-\mathrm{dry}\;\mathrm{weight}}{\mathrm{Fresh}\;\mathrm{weight}}\times 100$$

### 2.4. Sample Preparation and Preservation

The dry leaf samples were then ground to a fine powder using a 7010HG blender, Waring Commercial, Torrington, CT, USA. Methanol was used to extract TPC and TFC, and assess the antioxidant activity of powdered sweet potato leaves. For the antimicrobial tests, various solvents—acetone, methanol, ethanol, and hexane—were each employed at a powder-to-solvent ratio of 1:10 (*w*/*v*). Two extraction solvents were used for HPLC to extract phenolic compounds from sweet potato leaves. The first solution was 25 mL of methanol and 25% HCl in a 4:1 ratio. The second was 25 mL of pure methanol. Each solvent was mixed with 1 g of ground leaf. The sonic dismembrator/ultrasonicator (Fisherbrand™ Model 120 Sonic Dismembrator, Pittsburgh, PA, USA) was suspended in the solution for 20 min. The contents were transferred to a centrifuge tube and centrifuged at 3000 rotations per minute for 10 min. The resulting supernatant was collected in a falcon tube and refrigerated at −4 °C for subsequent analysis. The analytical procedures were performed in triplicate on the extracted samples to ensure reproducibility and accuracy of the results. The percentage yield of the extractive values from the four solvents—methanol, ethanol, acetone, and hexane—was tested using the following formula [17].
$$\mathrm{The}\;\mathrm{yield}\;\mathrm{of}\;\mathrm{extract}\;(\%)=\frac{\mathrm{Weight}\;\mathrm{of}\;\mathrm{solvent}-\mathrm{free}\;\mathrm{extract}\;\left(g\right)\;}{\mathrm{Weight}\;\mathrm{of}\;\mathrm{dried}\;\mathrm{extract}\;\left(g\right)}\times 100$$

### 2.5. Spectrophotometric Assay for Total Phenolic Content

The TPC of the extracts was evaluated through a spectrophotometric assay following the method outlined in our previous study [18].

### 2.6. Total Flavonoids Content

The TFC in the samples was determined using an aluminum chloride colorimetric method with modifications [19]. The modifications included using 0.1 mL of 10% AlCl_3_ instead of 0.2 mL. Absorbance readings were taken at a wavelength of 415 nm using a UV-Vis spectrophotometer (ASYS UVM 340 plate reader, Biochrom US, Holliston, MA, USA). The TFC was expressed in mg of Quercetin Equivalent per gram of Dry weight (mg QE/g Dw).

### 2.7. ABTS and DPPH Radical Scavenging Assay

In vitro antioxidant activity was assessed using the ABTS and DPPH assays described by Everette and Islam [20], and Nithianantham et al. [21]. The percentage of free radical scavenging was then calculated using the designated equation:$$\mathrm{Scavenging}\;\mathrm{effect}\;(\%)=\frac{\mathrm{Blank}\;\mathrm{absorbance}\;-\;\mathrm{sample}\;\mathrm{absorbance}\;}{\mathrm{Blank}\;\mathrm{absorbance}}\times 100$$
where blank absorbance refers to the absorbance of the control reaction (containing all reagents except the test samples).

### 2.8. FRAP Assay

This reducing power assay is another technique for assessing antioxidant capacity measured using a modified version of the method outlined by Liu et al. [22]. For the sample preparation, 50 µL of the plant extract was diluted to 1 mL with methanol. Trolox, at equivalent concentrations, was used as a positive control. The FRAP value was determined using the following equation:$$\mathrm{FRAP}\;\mathrm{value}=\frac{\mathrm{Sample}\;\mathrm{absorbance}\;-\;\mathrm{blank}\;\mathrm{absorbance}\;}{\mathrm{Absorbance}\;\mathrm{of}\;\mathrm{positive}\;\mathrm{control}\;-\;\mathrm{blank}\;\mathrm{absorbance}}\times 2$$

### 2.9. Antimicrobial Assay Determination

The Kirby–Bauer disk diffusion method [23] was employed to assess the antimicrobial properties of the extracts. This in vitro method evaluated the antimicrobial activity against Gram-positive bacteria (*S. aureus*, *S. epidermidis*, *S. mutans*, *L. monocytogenes*, and *B. subtilis*), Gram-negative bacteria (*E. coli*, *S. dysenteriae*, and *K. aerogenes*), and fungal strains (*C. albicans* and *A. niger*). Rifampin, azithromycin, vancomycin, and voriconazole were used as positive controls, while the solvent used for extraction was a negative control.

#### 2.9.1. Preparation of Inocula

MacConkey agar, Tryptic soy agar, Mueller Hinton Agar (MHA), and Sabouraud Dextrose Agar (SDA) were prepared according to the manufacturer’s instructions and sterilized by autoclaving. Once the media had solidified in Petri plates, bacterial and fungal strains from Kwik Stik and Culti-Loops were streaked onto specific agars: MacConkey Agar for Gram-negative bacteria; Tryptic Soy Agar for Gram-positive bacteria; and SDA for fungal species. Incubation was performed at 35 °C for 24 h, following which colonies were collected and resuspended in sterile 0.9% saline to achieve a standardized concentration of 1.5 × 10^8^ CFU/mL, corresponding to 0.5 McFarland standards. The standardized suspensions were then inoculated onto fresh media plates using sterile wire loops.

#### 2.9.2. Evaluation of Antimicrobial Activity

Circular paper discs of approximately 10 mm in diameter were created from Whatman No. 1 filter paper. These discs were impregnated with 2 mL of extract (10 mg/mL). After allowing the solvent to evaporate, the discs were applied to MHA plates pre-seeded with bacterial inocula for the antibacterial tests, and to SDA plates pre-seeded with fungal spores for antifungal tests. These plates were then incubated at 35 °C for 24 h. Post-incubation, the inhibition zones on the agar surface around the discs (including the diameter of discs) were measured with slide calipers, indicating the extract’s effectiveness against the test organisms by the clearances in millimeters observed on the agar surface.

### 2.10. High-Performance Liquid Chromatography (HPLC)

The chromatography used a Hitachi LaChrom Elite HPLC system with an L-2455 Diode Array Detector and a reversed-phase C18 column (4.6 I.D. × 150 mm, 5-µm). The mobile phase for CA analysis followed isocratic elution by Truong et al. [24]. In contrast, ChA, 3,4-diCQA, and 3,5-diCQA were separated using a modified method by Nour et al. [25] with a 30 min linear gradient. Quercetin and gallic acid were analyzed as described by Fernandes et al. [26]. Detection wavelengths were 325, 326, 360, and 271 nm, with a flow rate of 1.0 mL/min and a column temperature of 35 °C.

### 2.11. Statistical Analysis

The Shapiro–Wilk test was used to assess normality before conducting the comparison test. The independence of observations was ensured through the study design, and homogeneity of variance was confirmed using Levene’s Test. The data showed that all assumptions were met. In our study, all experimental results are expressed as the mean ± Standard Error of the Mean (SEM). For statistical comparisons, we applied Analysis of Variance (ANOVA) followed by Tukey’s multiple comparison test using SPSS software, Version 27.0 (IBM Corp., 2022, Armonk, NY, USA). Statistically significant differences were defined as those with a *p*-value of <0.05.

## 3. Results and Discussion

### 3.1. Phenotypic Characteristics

Table 1 shows the phenotypic traits of six *Ipomoea batatas* varieties. ‘Vardaman’ had the longest internode distance (8.33 ± 0.76 cm) and ‘Centennial’ had the shortest (3.10 ± 0.82 cm). ‘Georgia Jet’ had the longest petiole (16 ± 1.5 cm) and ‘Covington’ the shortest (6.93 ± 2.48 cm). ‘Centennial’ had the largest leaves, while ‘O’Henry’ had the smallest (Figure 1). Subtle genetic differences within a species and variations in microclimate conditions such as soil quality, sun exposure, and water availability, even within a small geographical area, can result in differences in physical traits like leaf size and shape among varieties of the same species. Furthermore, epigenetic changes can cause phenotypic variations with similar genetic structures. However, integrating phenotypic and genetic data is vital for analyzing the causes of trait variations within the same taxon and evaluating the adaptive potential of these populations in the future. Research indicates that plants with higher ploidy, or the degree of repetition in their chromosomes, are better equipped to adapt to difficult growing conditions due to their extra-genetic diversity. Measuring key adaptation is essential for selecting variety with superior agronomic performance, especially for growers aiming to produce high-performing crops [27].

### 3.2. Moisture Contents

Table 2 shows the moisture content of sweet potato leaves, ranging from 72.89% (O’Henry) to 84.14% (Centennial). Leaf moisture content may be affected by maturity. Hong et al. [28] found moisture content in 13 sweet potato cultivars ranged from 87.77 to 90.27 g/100 g FW. Water stress, including drought and hypersalinity, impairs plant function, growth, and photosynthesis, reducing crop yield. Drought and salinity-resistant varieties maintain consistent water content and higher dry matter [29], crucial for water-scarce regions. Leaf moisture content guides fertilization and irrigation strategies, making it pivotal for effective crop management.

### 3.3. Extraction Yield Efficiency of Various Solvents

Table 2 presents the extraction efficiency of various solvents. Extraction yield increased with solvent polarity, with methanol yielding significantly higher efficiencies than ethanol, acetone, or hexane. Arawande [30] investigated the extraction yields of sweet potato leaves using methanol, ethanol, acetone, chloroform, ethyl acetate, and aqueous solvents, which yielded 2.345, 6.057, 5.175, 2.065, 6.214, and 8.544%, respectively. Zhang et al. [31] reported a yield of 344.1 mg dried (crude) extract/g dried leaf using a 70% ethanol solution. Purifying the crude extract with different solvents resulted in the highest yield from the water fraction (219.8 mg/g), followed by the petroleum ether (111.2 mg/g) and ethyl acetate fractions (12.5 mg/g), indicating the presence of water-soluble components such as proteins, minerals, and carbohydrates in sweet potato leaves. Our study employed Ultrasound-Assisted Extraction (UAE), using high-frequency electromagnetic waves (20 to 2000 kHz) to enhance extraction through acoustic cavitation. This improves solvent–plant material contact and cell wall permeability, making UAE superior to other methods.

### 3.4. Evaluation of Total Phenolic Content (TFC) and Total Flavonoid Content (TFC)

Table 3 details the analysis of TPC and TFC in six sweet potato leaf varieties. The differences among these varieties were not statistically significant. The TPC was significantly higher than the TFC. ‘O’Henry’ had the highest phenol (10.50 ± 1.04 mg) and flavonoid (4.26 ± 0.23 mg) content, while ‘Centennial’ had the lowest TPC and TFC at 7.29 ± 0.62 mg and 2.30 ± 0.04 mg. Various factors, including genotype, environmental conditions, cultivation techniques, and harvesting period, influence the phenolic content in sweet potato leaves. This study highlights varietal differences as the primary driver of phenol content variation by maintaining consistent environmental conditions and cultivation practices. Comparative studies by Islam et al. [8,32] corroborate that sweet potato leaves contain higher phenolic values than other plant parts, such as storage roots or potato tubers. They also contain more phenolics than commonly consumed vegetables like spinach (3.68 mg GAE/g) and pumpkin leaves (3.60 mg TAE/g).

Phenolics and flavonoids are primary antioxidants; flavonoids, produced in response to microbial infections, donate electrons to neutralize free radicals. Research shows a strong correlation between a plant’s antioxidant abilities and its phenolic and flavonoid contents—Gong [33]. TPC and TFC of five varieties of Romanian sweet potato leaves varied between 120.53 and 40.13 mg GAE/100 g FW, and between 15.65 and 6.04 µg QE/100 g FW [34]. Additionally, TPC in the leaves of four South African sweet potato cultivars, a cultivar from Peru, and ‘Beauregard’ from the USA ranged from 2319.10 to 1322.76 mg/kg [35], which is lower than the TPC found in our study. Our study’s TFC exceeds those found in Ghasemzadeh et al. [36], where levels ranged from 1.87 ± 0.84 to 3.95 ± 0.91 mg QE/g. However, comparing these data is challenging due to variations in extraction techniques, solvents, and analytical methods used across studies.

### 3.5. Antioxidant Activity (AA) of Sweet Potato Leaves

Our study assessed the AA of six sweet potato cultivars using ABTS, DPPH, and FRAP assays, comparing results with standard antioxidants Trolox (ABTS, FRAP) and ascorbic acid (DPPH) in Table 4. In the ABTS assay, ‘O’Henry’ had the highest radical scavenging activity (0.81 ± 0.30 mg), followed by ‘Vardaman’, and ‘Covington’ had the lowest (0.66 ± 0.82 mg). DPPH assay activity ranged from 110.05 to 125.04 mg; it was highest in ‘O’Henry’ and lowest in *‘*Georgia Jet’. The FRAP assay showed the highest antioxidant potential in O’Henry (2.83 ± 0.07) and the lowest in Beauregard (1.98 ± 0.14). The IC_50_ value quantifies the extract concentration needed to neutralize 50% free radicals, inversely reflecting scavenging capacity. Table 4 provides IC_50_ values for DPPH and ABTS assays, with ‘O’Henry’ displaying the lowest IC_50_ and superior antioxidant activity. Figure 2 illustrates scavenging activities, consistent with ‘O’Henry’s high total polyphenol content, affirming that polyphenols contribute significantly to radical scavenging abilities.

Pearson correlation coefficients (r) and coefficients of determination (R2) were conducted to explore the relationships between TPC, TFC, and AA, as measured by FRAP, DPPH, and ABTS assays. The graphical representation of Figure 3 presents scatter plots illustrating these linear relationships. These analyses aimed to elucidate the contributions of phenolic compounds, specifically phenolic acids and flavonoids, to the antioxidant capacities of the leaves. A low positive correlation was observed between TFC and FRAP (r = 0.37, R^2^ = 0.14), as indicated in Figure 3(C2). A moderate positive correlation was noted between TFC and DPPH (r = 0.69, R^2^ = 0.47), shown in Figure 3(B2). A similar moderate positive correlation was seen between TFC and ABTS (r = 0.64, R^2^ = 0.41), highlighted in Figure 3(A2). TPC also exhibited a moderate positive correlation with ABTS (r = 0.62, R^2^ = 0.38), displayed in Figure 3(A1). Strong positive correlations were recorded between TPC and FRAP (r = 0.89, R^2^ = 0.80), as illustrated in Figure 3(C1). Additionally, a very strong positive correlation was found between TPC and DPPH (r = 0.93, R^2^ = 0.87), presented in Figure 3(B1).

The ABTS, DPPH, and FRAP antioxidant assays demonstrated significant variations in AA across cultivars, indicating genotype-dependent differences in radical scavenging abilities [37]. These differences are likely due to variations in TPC, types of polyphenols, and overall nutrient profiles. Sun et al. [38] indicated that sweet potato leaves from different cultivars vary not only in their phenolic constituents but also in the proportions of these constituents. Some polyphenols may not function as antioxidants, and their presence alongside antioxidant polyphenols within extracts could significantly impact overall antioxidant effectiveness. Moreover, the leaves differ in their proximate composition, including ash, crude fat, protein, fiber, and carbohydrate contents, which can synergistically or antagonistically influence the antioxidant effects of the polyphenols. A study by Suárez et al. [39] highlighted the importance of leaf harvest timing, suggesting late September is optimal for achieving the highest phenolic content, directly correlating with increased antioxidant activity.

The antioxidant capacities documented in this study surpass those found in earlier studies by Ghasemzadeh et al. [36]. Specifically, Ghasemzadeh et al. [34] documented higher IC_50_ values for the DPPH assay, indicating lesser antioxidant activity. Their results showed IC_50_ values of 184.30 μg/mL for Vardaman, 381.20 for Georgia Jet, 226.92 for Beauregard, and 450.46 for Centennial, significantly higher than our present study. Phahlane et al. [33] reported that the IC_50_ values for the Beauregard variety were 4.22 mg/mL for the DPPH assay and 3.54 mg/mL for the ABTS assay. Yang et al. [40] also corroborated this high activity level and demonstrated that *Ipomoea batatas* had the highest DPPH radical scavenging activity among 23 vegetable species commonly consumed in Taiwan.

### 3.6. Outcomes of the Antimicrobial Assay Analysis

Table 5 displays the antibacterial and antifungal efficacy of leaf extracts from six varieties of *I. batatas*, based on the solvent used for extraction, quantified by the zone of inhibition in millimeters. Methanol extracts from most varieties were effective against *S. aureus*, with ‘Vardaman’ showing the highest inhibition (19.33 mm), comparable to the positive control vancomycin (20 mm). Many varieties, particularly methanol and ethanol extracts, showed activity against *S. mutans*, with ‘Vardaman’s methanol extract being the most potent (32.33 mm), though lower than positive controls. Methanol and acetone extract from most varieties showed some activity against *L. monocytogenes*. None of the extracts inhibited *S. epidermidis* or *B. subtilis*. Extracts displayed varying activity against *E. coli*, *S. dysenteriae*, and *K. aerogenes*, with strong activity from ‘Covington’ (22.33 mm) and ‘O’Henry’ (19.67 mm) methanol extracts against *E. coli*, surpassing the positive control rifampin (13.67 mm). ‘Covington’ (20.50 mm) acetone extract and ‘O’Henry’ (19.83 mm) methanol extract showed higher inhibition against *S. dysenteriae* compared to rifampin (14.67 mm).

Several varieties displayed moderate antifungal activity against *C. albicans*, with ‘Covington’ and ‘O’Henry’ ethanol extracts, and ‘Vardaman’s acetone extract, being most effective (20.83, 18.33, 19.17 mm). Activity against *A. niger* was notable in ‘Vardaman’s hexane, ‘O’Henry’s methanol, and ‘Beauregard’s ethanol extract, with inhibition zones of 24.67, 17.67, and 19 mm. These results are lower than the positive control voriconazole. Adsull et al. [41] found that acetone and ethanol extracts of sweet potato leaves exhibited antimicrobial effects against *Salmonella* typhimurium and *Pseudomonas aeruginosa*. Mbaeyi-Nwaoha and Emejulu [42] demonstrated significant antimicrobial activity of water extracts against *S. typhi*, *S. aureus*, *A. niger*, *P. aeruginosa*, and *K. pneumoniae*, but not against *E. coli*. Conversely, Pochapski et al. [43] reported that freeze-dried sweet potato extracts in 70% ethanol showed no growth inhibition against *S. mutans*, *S. aureus*, and *C. albicans*.

Greenwood [44] classified microbial sensitivity based on inhibition zones: strong (>20 mm), medium (16–20 mm), weak (10–15 mm), and no response (<5 mm). Our study shows methanol and ethanol extracts had higher antimicrobial activity, especially against Gram-positive bacteria and fungi, while hexane showed the least activity. These observations may be attributed to methanol’s higher polarity, which allows it to extract more active antibacterial constituents. Gram-positive bacteria were more susceptible due to their simpler cell wall structure. In contrast, Gram-negative bacteria’s complex outer membrane with a hydrophilic surface and enzymes in the periplasmic space that can degrade foreign molecules hindered antimicrobial penetration and efficacy [45]. The antimicrobial activity is due to secondary metabolites like phenolic compounds present in the extracts. Polyphenols contain polar hydroxyl groups that effectively bind to bacteria. They are considered weak organic acids, are lipophilic, penetrate microbial membranes via passive diffusion, disrupt membrane integrity, acidify the cytoplasm, and cause microbial cell death by leaking essential intracellular components [46].

Further supporting this mechanism, Lou et al. [47] demonstrated that *p*-coumaric acid, a specific phenolic compound, exerts antibacterial effects by disrupting cell membranes and binding to their DNA. Sweet potato leaves contain phenolic compounds like caffeic acid derivatives, rutin, kaempferol, myricetin, *p*-coumaric acid, apigenin, and luteolin [8,9], which may collectively inhibit microbial growth and antibiotic resistance mechanisms. Despite the low antibacterial activity of extracts, the presence of these single purified compounds suggests potential antimicrobial properties. This highlights the need for thorough future research to identify and characterize active antibacterial compounds.

### 3.7. RP-HPLC-DAD Analysis

#### 3.7.1. Specificity

The specificity evaluation involved comparing chromatograms from blank, standard, and sample solutions. The sample chromatogram’s peak integration matched the standard. No additional peaks were observed within the retention time for the compounds of interest, suggesting no interference from the solvents. This demonstrates the HPLC method’s specificity for qualitative and quantitative analysis of phenolic compounds.

#### 3.7.2. HPLC Method Validation

The validation process addressed linearity, Limits of Detection (LOD), and Limits of Quantification (LOQ).

##### Linearity

Linearity was assessed using standard concentrations from 0.78 to 100 µg/mL, showing good linearity for the targeted compounds. HPLC analyses constructed the calibration curve by plotting peak area versus concentration on three days, with correlation coefficients always greater than 0.99. Phenolic compound content was determined by a regression equation using the calibration curve.

##### Limits of Detection (LOD) and Limits of Quantification (LOQ)

LOD and LOQ for each compound were determined through regression analysis. The minimal detectable analyte concentration was LOD = 3.3 × D/S. The lowest concentration at which the analyte produces a measurable response that can be precisely quantified was LOQ = 10 × D/S. Here, D is the standard deviation of the y-intercepts of the regression lines, and S denotes the slope of the calibration curve. The LOD and LOQ values for the HPLC method validation are as follows: CA, ChA, 3,5-diCQA, 3,4-diCQA, quercetin, and gallic acid had LODs of 3.67, 0.87, 1.27, 3.27, 1.30, and 1.68 µg/mL, respectively, and LOQs of 11.12, 2.64, 3.86, 9.90, 3.93, and 5.08 µg/mL, respectively.

#### 3.7.3. Optimization of Extraction Efficacy

The extraction efficiency of CA, ChA, 3,4-diCQA, 3,5-diCQA, quercetin, and gallic acid from sweet potato leaves was evaluated using acidified methanol and pure methanol as solvents. Acidified methanol was the most effective, yielding the highest concentrations of the target compounds. Samples extracted with pure methanol had significantly lower levels of all compounds, as shown in Table 6.

#### 3.7.4. Analysis of Phenolic Compounds in Sweet Potato Leaf

The chromatograms of standard compounds and sweet potato leaf extracts are shown in Figure 4. Based on HPLC Retention Times (RT) and UV absorbance spectra, the compounds were tentatively identified as CA (peak 1 at RT 3.16 min), ChA (peak 2 at RT 9.70 min), 3,5-diCQA (peak 3 at RT 17.76 min), 3,4-diCQA (peak 4 at RT 20.36 min), and quercetin (peak 5 at RT 18.66 min). Gallic acid was not detected in any sample extracts. The contents of these compounds are presented in Table 6. The results of the phenolic acid and flavonoid contents revealed that 3,5-diCQA was the most abundant among the six compounds. Caffeic acid content in acidified methanol ranged from 0.31 to 1.25 mg/g, chlorogenic acid from 2.24 to 6.56 mg/g, 3,5-diCQA from 9.91 to 21.80 mg/g, and 3,4-diCQA from 0.82 to 2.18 mg/g. Significant differences were observed among cultivars. ‘O’Henry’, ‘Covington’, and ‘Centennial’ contained the highest levels of chlorogenic acid and 3,4-diCQA, with ‘Centennial’ having the maximum level of 3,5-diCQA, followed by ‘Covington’ and ‘O’Henry’. Regarding caffeic acid, ‘O’Henry’ and ‘Covington’ had the highest levels. Quercetin, a key flavonoid, was present in lower concentrations in all cultivars except ‘O’Henry’, with levels ranging from 0.04 to 0.21 mg/g Dw.

Previous studies have identified phenolic acids and flavonoids as sweet potato’s primary classes of phenolic compounds. Islam et al. [8] isolated six polyphenols (3,5-diCQA, 4,5-diCQA, ChA, 3,4-diCQA, 3,4,5-triCQA, and CA) from the leaves of 20 sweet potato genotypes, with 3,5-diCQA being the most abundant (1062.34–1619.28 mg/100 g). Takenaka et al. [48] identified six polyphenols in raw sweet potato roots: β-d-fructofuranosyl 6-O-caffeoyl-α-d-glucopyranoside, CA, and four CQAs (3-CQA, 4-CQA, 5-CQA, 3,4-diCQA, 3,5-diCQA, and 4,5-diCQA), and two (3-CQA and 4-CQA) in heated sweet potatoes. In fresh roots, 5-CQA and 3,5-diCQA were predominant, while FCG was predominant in long-stored roots. Truong et al. [24] quantified five polyphenolic compounds (CA, ChA, 4,5-diCQA, 3,5-diCQA, and 3,4-diCQA) in the leaves and root tissue of three commercial cultivars in the U.S., finding that 3,5-diCQA was the main phenolic compound. The concentrations ranged as follows: 29.20 to 30.70, 683.00 to 814.90, 691.20 to 743.90, 821.30 to 851.80, and 101.70 to 218.80 mg ChAE/100 g dry weight. Dicaffeoylquinic acid has been suggested as having anti-adipogenesis, neuroprotective, anti-hyperglycemia, anti-mutagenic, and anti-melanogenic effects [49]. Flavonoids, including quercetin, are significant in the Convolvulaceae family’s biochemistry, including sweet potato. In all ten accessions analyzed, quercetin was the most abundant flavonoid in sweet potato’s leaf and storage root, comprising 0.22–1.34% of Dw [50]. Krochmal-Marczak et al. [51] showed that quercetin content in sweet potato leaves varied with growth stage and cultivar genotype, with higher concentrations at full ripeness, ranging from 413.20 mg/100 g DM to 1086.00 mg/100 g DM.

## 4. Conclusions

This study highlights the significant biological activities of *Ipomoea batatas* leaf extracts, with methanolic extracts proving the most effective. Acidified methanolic extracts were found to be rich in CA, chlorogenic acid isomers (ChA, 3,5-diCQA, 3,4-diCQA), and quercetin, and demonstrated notable in vitro antioxidant and antimicrobial activities. Both methanolic and ethanolic extracts showed bactericidal solid and fungicidal effects, particularly against Gram-positive bacteria against pathogens such as *S. aureus*, *S. mutans*, *L. monocytogenes*, *E. coli*, *S. dysenteriae*, and *C. albicans*, with methanolic extracts being the most effective. Among the tested cultivars, ‘O Henry’ exhibited the highest levels of total polyphenols, flavonoids, and antioxidant activity, suggesting its potential as a functional food in the industry. The economic potential of utilizing sweet potato leaves as a source of natural antioxidants is promising, especially for communities with limited access to other sources. Thus, this study underscores the potential health benefits and the underutilized value of sweet potato leaves as a source of natural antioxidants and antimicrobials. Future research should focus on in vivo studies and further exploration of these extracts in the agri-food and pharmaceutical industries, emphasizing developing natural antibiotics to address antibiotic-resistant strains. Additionally, isolating other active components in pure form for potential use in infusions, nutraceuticals, and pharmaceuticals could make this plant a valuable candidate for managing infections, oxidative stress, and related disorders.

## Figures and Tables

**Figure 1 foods-13-02787-f001:**
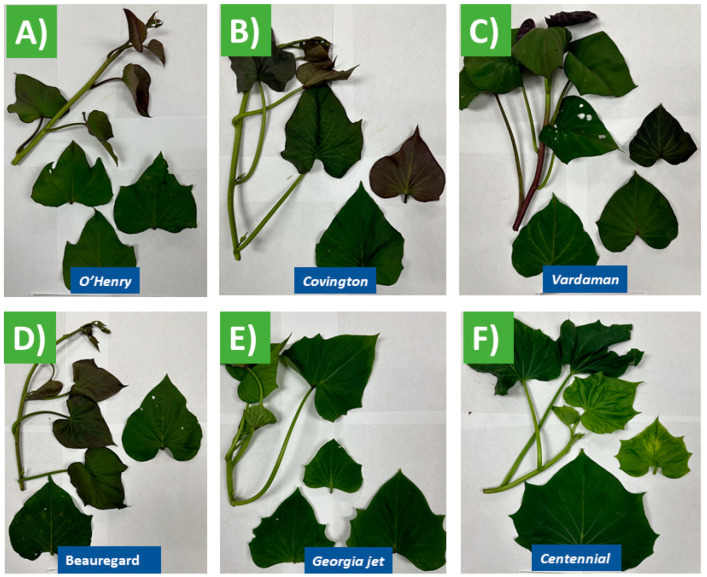
Leaves of (**A**) O’Henry, (**B**) Covington, (**C**) Vardaman, (**D**) Beauregard, (**E**) Georgia jet, (**F**) Centennial.

**Figure 2 foods-13-02787-f002:**
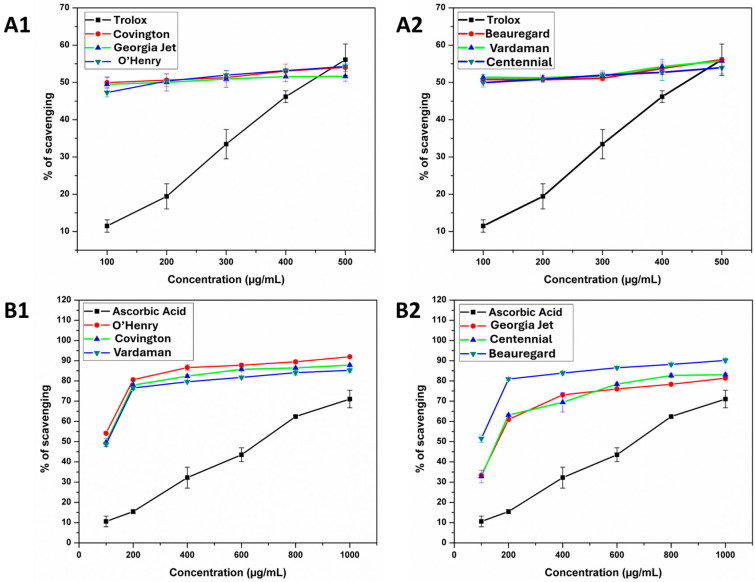
(**A1**,**A2**) ABTS, and (**B1**,**B2**) DPPH free radical scavenging activity of leaf extract of *Ipomoea batatas*. Each value is expressed in mean (*n* = 3) ± SEM.

**Figure 3 foods-13-02787-f003:**
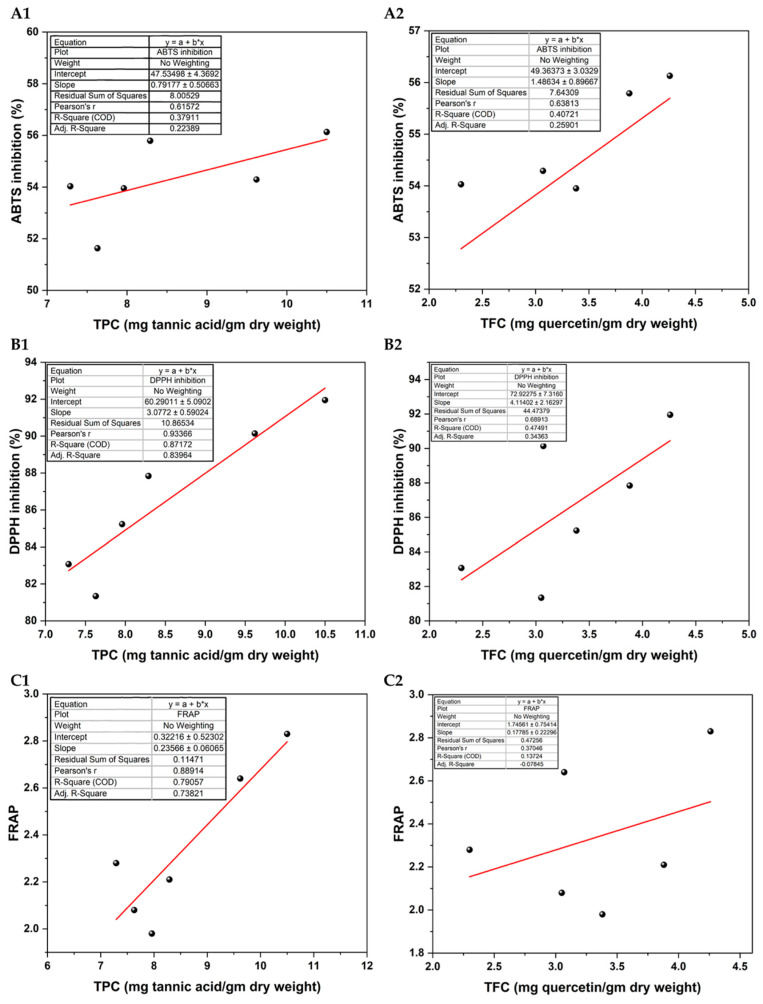
Correlation between total polyphenol content (**A1**,**B1**,**C1**) and total flavonoid content (**A2**,**B2**,**C2**) with antioxidant assays ABTS (**A1**,**A2**), DPPH (**B1**,**B2**), and FRAP (**C1**,**C2**). Each value is expressed in mean (*n* = 3) ± SEM.

**Figure 4 foods-13-02787-f004:**
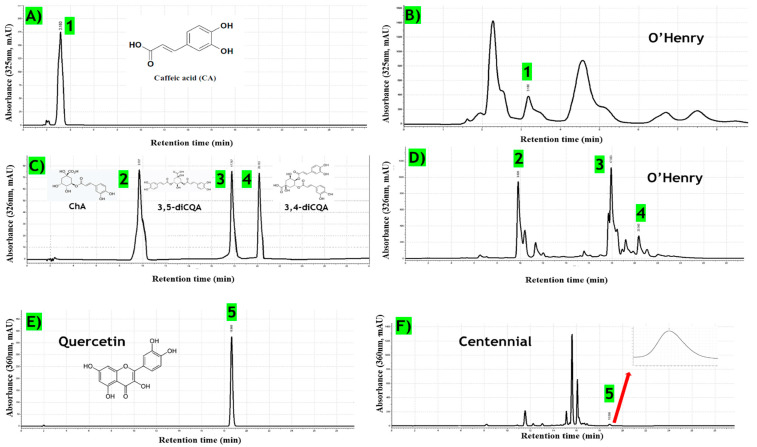
HPLC chromatogram of standards and acidified methanol extract of leaves. (**A**) standard: CA, (**B**) sample: O’Henry, (**C**) standard: ChA, 3,5-diCQA, and 3,4-diCQA, (**D**) sample: O’Henry, (**E**) standard: quercetin, (**F**) Sample: Centennial.

**Table 1 foods-13-02787-t001:** Phenotypic Characteristics of the leaves of *Ipomoea batatas*.

Varieties	Internode Distance (cm)	Petiole Length (cm)	Leaf Length (cm)	Leaf Width (cm)
O’Henry	7.37 ± 0.78 ^a^	14.00 ± 2.00 ^a^	6.00 ± 0.44 ^c^	6.63 ± 0.60 ^d^
Covington	4.07 ± 0.40 ^b^	6.93 ± 2.48 ^b^	8.63 ± 0.40 ^b^	10.50 ± 0.50 ^b,c^
Vardaman	8.33 ± 0.76 ^a^	15.33 ± 1.04 ^a^	9.33 ± 1.26 ^b^	12.77 ± 1.50 ^a,b^
Beauregard	4.17 ± 0.76 ^b^	8.50 ± 2.18 ^b^	8.50 ± 0.87 ^b^	8.33 ± 0.58 ^c,d^
Georgia Jet	3.83 ± 0.76 ^b^	16.00 ± 1.50 ^a^	10.50 ± 1.00 ^b^	11.67 ± 2.52 ^a,b,c^
Centennial	3.10 ± 0.82 ^b^	13.60 ± 0.53 ^a^	12.73 ± 0.38 ^a^	14.20 ± 0.53 ^a^

The mean (*n* = 3) ± SEM followed by different lowercase letters indicates a significant difference (*p* < 0.05) among cultivars, as measured by the Tukey test.

**Table 2 foods-13-02787-t002:** Moisture content and extraction yield of leaf extracts of *Ipomoea batatas*.

Sample	Moisture Content (%)	Extraction Yield (%)			
		Methanol	Ethanol	Acetone	Hexane
O’Henry	72.89 ± 1.71 ^a^	18.79 ± 1.71 ^f^	12.31 ± 1.69 ^d,e^	6.46 ± 1.24 ^a,b,c,d^	3.25 ± 2.00 ^a,b,c^
Covington	80.00 ± 3.00 ^b,c^	18.19 ± 2.87 ^e,f^	9.49 ± 0.57 ^b,c,d^	5.61 ± 0.86 ^a,b,c^	3.81 ± 2.79 ^a,b,c^
Vardaman	81.40 ± 1.99 ^b,c^	16.94 ± 1.81 ^e,f^	8.85 ± 0.84 ^b,c,d^	5.90 ± 0.92 ^a,b,c,d^	2.25 ± 0.12 ^a^
Beauregard	77.31 ± 2.03 ^a,b^	18.46 ± 2.65 ^e,f^	9.33 ± 1.41 ^b,c,d^	5.42 ± 0.35 ^a,b,c^	3.06 ± 1.49 ^a,b^
Georgia jet	79.29 ± 1.16 ^b,c^	18.44 ± 2.51 ^e,f^	9.63 ± 1.47 ^c,d^	6.25 ± 0.59 ^a,b,c,d^	3.04 ± 1.36 ^a,b^
Centennial	84.14 ± 2.43 ^c^	16.68 ± 1.55 ^e,f^	8.69 ± 1.69 ^b,c,d^	5.21 ± 0.29 ^a,b,c^	1.81 ± 0.04 ^a^

The mean (*n* = 3) ± SEM followed by different lowercase letters indicates a significant difference (*p* < 0.05) among cultivars, as measured by the Tukey test.

**Table 3 foods-13-02787-t003:** Total phenolic and total flavonoid contents of leaf extracts of *Ipomoea batatas*.

Varieties	Total Phenol (mg TAE/g Dw)	Total Flavonoid (mg QE/g Dw)
O’Henry	10.50 ± 1.04 ^a^	4.26 ± 0.23 ^a^
Covington	9.62 ± 0.80 ^a^	3.07 ± 1.02 ^a^
Vardaman	8.29 ± 0.83 ^a^	3.88 ± 0.13 ^a^
Beauregard	7.96 ± 0.89 ^a^	3.38 ± 0.12 ^a^
Georgia Jet	7.63 ± 0.49 ^a^	3.05 ± 0.14 ^a^
Centennial	7.29 ± 0.62 ^a^	2.30 ± 0.04 ^a^

The mean (*n* = 3) ± SEM followed by different lowercase letters indicates a significant difference (*p* < 0.05) among cultivars as measured by the Tukey test. TAE = Tannic Acid Equivalent, QE = Quercetin Equivalent, Dw = Dry weight.

**Table 4 foods-13-02787-t004:** Antioxidant activity of leaf extracts of *Ipomoea batatas*.

Varieties	ABTS Assay		DPPH Assay		FRAP Value
	Total Antioxidant Capacity (mg TE/g Dw)	IC_50_ Value (µg/mL)	Total Antioxidant Capacity (mg AAE/g Dw)	IC_50_ Value (µg/mL)	
O’Henry	0.81 ± 0.30 ^b^	88.80 ± 15.30 ^b^	125.04 ± 0.73 ^a^	88.83 ± 1.94 ^c^	2.83 ± 0.07 ^a^
Covington	0.66 ± 0.82 ^a^	115.17 ± 7.65 ^a^	122.56 ± 1.59 ^a^	95.17 ± 4.77 ^b,c^	2.64 ± 0.11 ^a^
Vardaman	0.80 ± 0.63 ^b^	94.60 ± 2.76 ^a,b^	119.40 ± 0.69 ^b^	99.50 ± 4.92 ^b,c^	2.21 ± 0.07 ^b,c^
Beauregard	0.78 ± 2.10 ^b^	100.47 ± 5.37 ^a,b^	115.83 ± 1.32 ^c^	110.67 ± 14.75 ^b^	1.98 ± 0.14 ^c^
Georgia Jet	0.67 ± 1.67 ^a^	104.33 ± 10.60 ^a,b^	110.05 ± 0.73 ^d^	147.60 ± 1.22 ^a^	2.08 ± 0.07 ^b,c^
Centennial	0.68 ± 2.08 ^a^	99.9 ± 6.50 ^a,b^	112.86 ± 0.93 ^d^	143.63 ± 2.80 ^a^	2.28 ± 0.07 ^b^

Mean (*n* = 3) ± SEM followed by different lowercase letters indicates significant difference (*p* < 0.05) among cultivars as measured by Tukey test. TE = Trolox Equivalent, AAE = Ascorbic Acid Equivalent.

**Table 5 foods-13-02787-t005:** Antimicrobial activity (zone of inhibition, mm) of leaf extracts of *Ipomoea batatas*.

Varieties	Extracted Solvent	*S. aureus*	*S. epidermidis*	*S. mutans*	*B. subtilis*	*L. monocytogenes*	*E. coli*	*S. dysentriae*	*K. aerogenes*	*C. albicans*	*A. niger*
O’Henry	Methanol	14.67 ± 1.53	-	21.67 ± 3.21	-	12.67 ± 2.08	19.67 ± 1.53	19.83 ± 1.53	-	-	17.67 ± 0.58
	Ethanol	-	-	21.33 ± 0.58	-	-	18 ± 1.00	-	-	18.33 ± 1.15	-
	Acetone	-	-	-	-	-	18.33 ± 0.58	18.17 ± 1.04	-	16.83 ± 1.26	-
	Hexane	-	-	12 ± 1.00	-	-	11.33 ± 0.58	-	-	-	-
Covington	Methanol	16.67 ± 1.53	-	21.33 ± 2.08	-	-	22.33 ± 2.52	-	-	11.5 ± 1.32	14.33 ± 2.08
	Ethanol	-	-	-	-	-	13.67 ± 1.53	-	-	20.83 ± 0.76	-
	Acetone	-	-	13.33 ± 1.53	-	13 ± 1.00	-	20.5 ± 1.50	-	-	-
	Hexane	-	-	-	-	-	11.33 ± 0.58	-	-	15 ± 3.60	-
Vardaman	Methanol	19.33 ± 0.58	-	32.33 ± 1.53	-	-	13 ± 1.00	-	-	13.5 ± 0.50	15.33 ± 3.51
	Ethanol	-	-	20.33 ± 0.58	-	-	12.67 ± 1.55	11.67 ± 0.58	-	10.83 ± 0.29	13.00 ± 0.58
	Acetone	-	-	-	-	12.67 ± 0.58	-	15.33 ± 1.53	-	19.17 ± 1.04	-
	Hexane	-	-	-	-	-	13.33 ± 0.58	-	-	11.17 ± 0.76	24.67 ± 3.51
Beauregard	Methanol	-	-	27.33 ± 1.53	-	12.33 ± 0.58	-	18.33 ± 0.76	-	-	15 ± 1.00
	Ethanol	-	-	16.33 ± 1.53	-	14 ± 1.00	11.33 ± 0.58	16.67 ± 1.15	-	17.33 ± 1.53	19 ± 3.06
	Acetone	-	-	-	-	11.33 ± 0.58	-	11.33 ± 0.58	-	12.5 ± 0.50	-
	Hexane	-	-	-	-	-	11.33 ± 0.58	-	-	-	-
Georgia Jet	Methanol	13.33 ± 0.58	-	26.67 ± 1.53	-	11.33 ± 0.58	-	-	-	-	13.33 ± 1.53
	Ethanol	-	-	18.67 ± 1.53	-	-	12.67 ± 1.55	-	-	13.33 ± 1.53	-
	Acetone	-	-	-	-	11.33 ± 0.58	-	-	-	-	-
	Hexane	-	-	-	-	-	10.83 ± 0.29	-	-	13.83 ± 0.76	-
Centennial	Methanol	13.67 ± 0.58	-	19.67 ± 1.15	-	-	-	-	-	-	17.33 ± 0.58
	Ethanol	-	-	-	-	-	-	12.33 ± 0.58	-	-	-
	Acetone	-	-	18.67 ± 1.53	-	-	-	18.03 ± 0.29	-	-	-
	Hexane	-	-	-	-	-	10.5 ± 0.50	-	-	-	-
Rifampin		30.00	38.67 ± 0.69	C/I	20.33 ± 1.53	30.33 ± 0.58	13.67 ± 1.53	14.67 ± 1.53	-	Voriconazole-C/I	Voriconazole- C/I
Azithromycin		-	-	C/I	28.67 ± 1.53	22 ± 1.00	24.33 ± 2.08	20 ± 2.00	9.33 ± 0.58		
Vancomycin		20.00	18 ± 1.00	C/I	20.33 ± 0.58	23.33 ± 1.53	-	12.33 ± 0.58	-		

Values are mean (3) ± SEM; - = no zone of inhibition; CI = Complete Inhibition.

**Table 6 foods-13-02787-t006:** Quantification (mg/gm dry weight) of caffeic acid, chlorogenic acid, 3,4-diCaffeoylquinic acid, 3,5-diCaffeoylquinic acid, and quercetin in different sweet potato cultivars.

Sample	Caffeic Acid		Chlorogenic Acid		3,5-Dicaffeoylquinic Acid		3,4-Dicaffeoylquinic Acid		Quercetin		Gallic Acid
	Acidified Methanol Extracted	Methanol Extracted	Acidified Methanol Extracted	Methanol Extracted	Acidified Methanol Extracted	Methanol Extracted	Acidified Methanol Extracted	Methanol Extracted	Acidified Methanol Extracted	Methanol Extracted	
O’Henry	1.25 ± 1.55 ^c^	0.99 ± 0.85 ^c^	6.56 ± 0.19 ^a^	2.88 ± 3.27 ^a^	14.35 ± 3.86 ^c^	8.49 ± 8.03 ^c^	2.18 ± 0.70 ^a^	1.50 ± 2.24 ^a^	ND	ND	ND
Covington	0.75 ± 0.99 ^b^	0.48 ± 0.69 ^b^	4.88 ± 6.72 ^b^	1.68 ± 1.34 ^b^	19.92 ± 8.38 ^b^	11.22 ± 6.87 ^b^	2.06 ± 1.87 ^a^	1.33 ± 0.56 ^b^	0.21 ± 1.74 ^a^		
Vardaman	0.74 ± 3.49 ^b^	0.49 ± 0.49 ^b^	2.24 ± 1.26 ^e^	1.27 ± 1.20 ^e^	9.95 ± 3.82 ^e^	5.33 ± 2.64 ^e^	1.08 ± 0.85 ^d^	1.03 ± 2.99 ^c^	0.08 ± 0.21 ^b^		
Beauregard	0.72 ± 6.89 ^b^	0.47 ± 0.07 ^b^	3.17 ± 5.82 ^d^	1.51 ± 1.42 ^c^	12.93 ± 6.35 ^d^	6.66 ± 10.22 ^d^	1.63 ± 2.36 ^c^	1.07 ± 0.40 ^c^	0.06 ± 0.80 ^b^		
Georgia jet	0.31 ± 0.77 ^a^	0.21 ± 3.81 ^a^	3.04 ± 5.95 ^d^	1.40 ± 2.30 ^d^	9.91 ± 7.16 ^e^	4.41 ± 3.70 ^f^	0.82 ± 4.56 ^e^	0.71 ± 0.77 ^d^	0.04 ± 0.61 ^b^		
Centennial	0.64 ± 0.96 ^b^	0.46 ± 2.19 ^b^	3.55 ± 4.07 ^c^	1.65 ± 0.94 ^b^	21.80 ± 13.11 ^a^	12.47 ± 11.49 ^a^	1.81 ± 4.90 ^b^	1.30 ± 3.25 ^b^	0.20 ± 0.49 ^a^		

Mean ± SEM followed by different lowercase letters indicates a significant difference (*p* < 0.05) among cultivars, as measured by the Tukey test; ND—not detected.

## Data Availability

The data presented in this study are available on request from the corresponding author.

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
