# Peer review of "Phytochemical Profiling and Antimicrobial Properties of Various Sweet Potato (Ipomoea batatas L.) Leaves Assessed by RP-HPLC-DAD"

_foods, 2024, doi:10.3390/foods13172787_

Round 1

Reviewer 1 Report

Comments and Suggestions for Authors

This study investigated the leaves of six cultivars of Ipomoea batatas L. and focused on their total polyphenol content (TPC), total flavonoid content (TFC), antioxidant, and antimicrobial activities. The authors have carried out relatively well-developed experiments to demonstrate their ideas. However, some issues need to be addressed before acceptance: 

1. The extraction of active ingredients from plants is a complex process, and different processes may obtain different substancesabstract section lacks a description of the experimental data to enhance the reasonableness of the exposition. For this reason, the author's title seems to be rather large, and it is recommended to narrow down the scope to correspond to the content.

2. For Sample Preparation and Preservation, the author should describe the extraction process in more detail, such as temperature, etc.

3. The author mentioned powder-to-solvent ratio of 1:10 (w/v), why is this ratio set here? Have other ratios been considered?

4. The quality of Figure 3 is very average, suggest the author to improve.

5. The introduction should end with some additional relevant experiments done.

6. The conclusion, although the authors gave very constructive suggestions, is more general, vague and should be explored in future experiments.

7. Check that reference formatting is in line with journal requirements and grammar cheques are carried out throughout the text.

Author Response

Response to the comment of Reviewer 1:

Comment: The abbreviation number is separated from the surname in some and not in others, homogenize according to the instructions of the journal.

Reply: We have carefully reviewed the entire manuscript and made the necessary corrections to ensure consistency in the use of abbreviations and the formatting of terms, as per your suggestions. (Highlighted with red color)

Comment: In section 2.1. What were the environmental conditions of the crops, in addition to altitude and latitude?

Reply: I have added detailed information about the environmental conditions, including altitude, latitude, and other relevant factors in Section 2.1. (Please see Section 2.1, Highlighted with red color)

Comment: In section 2.2. the ® symbol must be superscript

Reply: Revised accordingly.

Comment: The formula on line 11 must be centered and the same size and font as the rest of the formulas in the manuscript.

Reply: The formula on line 11 has been centered and adjusted to match the size and font of the other formulas in the manuscript. (Highlighted with red color)

Comment: Line 141: the number of atoms is subscripted.

Reply: Revised accordingly.

Comment: The word “in vitro” does not have a hyphen.

Reply: Revised accordingly. (Highlighted with red color)

Comment: Line 174: ^8 must be superscripted, since in other scientific notations it is written without the power symbol. Homogenize where necessary.

Reply: Revised accordingly. (Highlighted with red color)

Comment: Line 178: there is no hyphen between “Whatman-No.1” and the numerical value “1” is separated from “No.”

Reply: Revised accordingly. (Highlighted with red color)

Comment: Section 2.9. Chromatography must be in full HPLC.

Reply: Revised accordingly. (Highlighted with red color)

Comment: Line 190: it is not “30-minute”, the correct form is “30 minute”

Reply: Revised accordingly. (Highlighted with red color)

Comment: In section 2.10. Statistical . . . , the version, year and location of the statistical program are missing. It is also not mentioned whether your data meets the statistical assumptions.

Reply: We have added the version, year, and location of the statistical program used. The Shapiro-Wilk test was used to assess normality, Levene’s Test confirmed homogeneity of variance, and all assumptions were met. (Highlighted with red color)

Comment: Lines 213 to 220: the text of the paragraph must be justified.

Reply: Revised accordingly.

Comment: Homogenize throughout the text, in the percentage values ​​to two or three digits after the period.

Reply: Percentage values throughout the text have been homogenized to two digits after the decimal point.

Comment: Table 3. “Total flavonoid” should be capitalized.

Reply: Revised accordingly. (Highlighted with red color)

Comment: Table 1. “Internode distance” is not understood, and the values ​​should be homogenized with one or two digits after the period (Review and adjust throughout the manuscript).

Reply: Revised accordingly (Highlighted with red color).

Comment: Line 236: Correct form; (20 to 2000 kHz)

Reply: Revised accordingly (Line 264) (Highlighted with red color)

Comment: The value of “50” should be subscripted “IC50”, review and homogenize throughout the manuscript.

Reply: The value "50" in “IC50” has been subscripted and homogenized throughout the manuscript.

Comment: Figure 2. If you comment on concentration, then the “X” axis is missing the volume units.

Reply: The "X" axis now included the appropriate volume units.

Comment: Line 358: “p” in “p-coumaric acid” is in italics.

Reply: Revised accordingly.

Comment: Table 6. Specify in the title bar “Caffeic acid, CA”

Reply: The title bar now specified Caffeic acid, CA.

Comment: Figure 3. There are too many retention times, only enable those related to the compounds to be identified.

Reply: Retention times have been reduced to only those relevant to the identified compounds.

Reviewer 2 Report

Comments and Suggestions for Authors

The abbreviation number is separated from the surname in some and not in others, homogenize according to the instructions of the journal.

Incorrect form: total polyphenol content (TPC), total flavonoid content (TFC), correct form: Total Polyphenol Content (TPC), Total Flavonoid Content (TFC)

Incorrect form: Ferric ion-reducing antioxidant potential (FRAP), Correct form: Ferric ion-Reducing Antioxidant Potential (FRAP), homogenize throughout the text so that the first letters are capitalized at the beginning, since the same pattern also exists in other abbreviations.

In section 2.1. What were the environmental conditions of the crops, in addition to altitude and latitude?

In section 2.2. the ® symbol must be superscript

The formula on line 11 must be centered and the same size and font as the rest of the formulas in the manuscript

Line 141: the number of atoms is subscripted

The word “in vitro” does not have a hyphen

Line 174: ^8 must be superscripted, since in other scientific notations it is written without the power symbol. Homogenize where necessary.

Line 178: there is no hyphen between “Whatman-No.1” and the numerical value “1” is separated from “No.”

Section 2.9. Chromatography must be in full HPLC

Line 190: it is not “30-minute”, the correct form is “30 minute”

In section 2.10. Statistical . . . , the version, year and location of the statistical program are missing. It is also not mentioned whether your data meets the statistical assumptions.

Lines 213 to 220: the text of the paragraph must be justified

Homogenize throughout the text, in the percentage values ​​to two or three digits after the period

Table 3. “Total flavonoid” should be capitalized

Table 1. “Internode distance” is not understood, and the values ​​should be homogenized with one or two digits after the period (Review and adjust throughout the manuscript).

Line 236: Correct form; (20 to 2000 kHz)

The value of “50” should be subscripted “IC50”, review and homogenize throughout the manuscript

Figure 2. If you comment on concentration, then the “X” axis is missing the volume units.

Line 358: “p” in “p-coumaric acid” is in italics

Table 6. Specify in the title bar “Caffeic acid, CA”

Figure 3. There are too many retention times, only enable those related to the compounds to be identified.

Author Response

Response to the comment of Reviewer 2:

  1. Suggest adding a summary and overview of the functional activities of sweet potato leaves in the preface, such as antioxidant activity, immune activity, etc.

Reply: As per the reviewer’s suggestion, we have added a summary and overview of the functional activities of sweet potato leaves, including their antioxidant and immune activities, in the Introduction Section. (Lines 73-87, Highlighted with red color).

  1. The phenotype characteristic results indicate that due to the samples being planted in the same place, it is necessary to discuss the significant differences in phenotype characteristics caused by genetic differences.

Reply: As per the reviewer’s suggestion, we have expanded the discussion in Section 3.1 to explain the contributions of genetic differences and environmental factors to the phenotypic variation observed among the different varieties of sweet potato plants despite them being planted in the same location (Line 222-233, Highlighted with red color).

  1. Different extraction reagents will inevitably impact the sample's extraction efficiency. The author discusses which reagent is more effective for analysis and easier to use for comparative analysis.

Reply: The authors agreed with the reviewer’s comment.

  1. In Fig.2, the reviewer did not find that the author used a positive control (Vc) in the antioxidant analysis. Please explain the reason for this.

Reply: We have included the positive control’s (Vc) scavenging activity in Figure 2, with Trolox used for the ABTS assay and Ascorbic Acid for the DPPH assay.

  1. The author needs to analyze and compare the correlation between the content of phytochemicals (polyphenols and flavonoids) in sweet potato leaves and the differences in antioxidant and antibacterial activities.

Reply: As per the reviewer’s suggestion, we have added an analysis and comparison of the correlation between the Total Polyphenol Content and Total Flavonoid Content with the antioxidant assays (ABTS, DPPH, and FRAP) in the manuscript, presented in Figure 3. (Figure 3 and Line 318-331, Highlighted with red color)

  1.  The reference format needs to be standardized, and it is recommended that DOI be added.

Reply: We have standardized the reference format throughout the manuscript and added DOI numbers to all references.

Reviewer 3 Report

Comments and Suggestions for Authors

1.       Suggest adding a summary and overview of the functional activities of sweet potato leaves in the preface, such as antioxidant activity, immune activity, etc.

2.       The phenotype characteristic results indicate that due to the samples being planted in the same place, it is necessary to discuss the significant differences in phenotype characteristics caused by genetic differences.

3.       Different extraction reagents will inevitably have a certain impact on the extraction efficiency of the sample. The author discusses which extraction reagent is more effective for analysis and is easier to use for comparative analysis.

4.       In Fig.2 In the antioxidant analysis, the reviewer did not find that the author used a positive control (Vc). Please explain the reason for this.

5.       The author needs to analyze and compare the correlation between the content of phytochemicals (polyphenols and flavonoids) in sweet potato leaves and the differences in antioxidant and antibacterial activities.

6.       The reference format needs to be standardized, and it is recommended to add DOI.

Comments on the Quality of English Language

None

Author Response

(The authors gave the same response as above.)

Round 2

Reviewer 2 Report

Comments and Suggestions for Authors

The number corresponding to the authors' affiliation is separated from the surname in some cases and together with the surname in others. Homogenize.

In line 97 and 98, the form is incorrect (Caffeic Acid (CA), Chlorogenic Acid (ChA), 3,4-diCaffeoylquinic Acid (3,4-diCQA), 3,5-diCaffeoylquinic Acid (3,5-), the correct form (CA, ChA, 3,4-diCQA, 3,5- . . ., since it has already been described in an abbreviated form, it is only the first time it is cited.

The coordinates on line 109 do not guide me to any place on Google Maps, are they written correctly? I know that it describes “University of Arkansas”, but the coordinates do not guide me.

On line 112: I suggest changing from: 92°F to 72°F to the following: 92 to 72°F

The formulas on lines 135, 152, 166, 173, still do not have the same font size. Also, remove bold in the formulas.

The equations on lines 166 and 173 are missing the name of the variable to be calculated (see equations 135 and 152 for more reference).

On line 258, the “%” symbol is only placed on the last value “8.544%”

Table 3. Suggested form “Total Flavonoid”

In figure 3, the box on the upper left margin of each figure is not understandable, improve the pixels.

Carefully review figure 4, the title of the “Y” axis is cut off in some figures, as well as the chemical structures, I suggest that they all be the same visible size and sharpness.

Author Response

 Comments and Suggestions for Authors

The number corresponding to the authors' affiliation is separated from the surname in some cases and together with the surname in others. Homogenize.

Ans: We have revised the manuscript, accordingly, ensuring that the full names of the compounds are only used upon first mention, with abbreviations used consistently thereafter.

In line 97 and 98, the form is incorrect (Caffeic Acid (CA), Chlorogenic Acid (ChA), 3,4-diCaffeoylquinic Acid (3,4-diCQA), 3,5-diCaffeoylquinic Acid (3,5-), the correct form (CA, ChA, 3,4-diCQA, 3,5- . . ., since it has already been described in an abbreviated form, it is only the first time it is cited.

Ans: Corrected

The coordinates on line 109 do not guide me to any place on Google Maps, are they written correctly? I know that it describes “University of Arkansas”, but the coordinates do not guide me.

Ans: We have thoroughly checked and corrected the coordinates in line 109 to ensure they accurately guide to the intended location at the University of Arkansas on Google Maps.

On line 112: I suggest changing from: 92°F to 72°F to the following: 92 to 72°F

Ans: Corrected the temperature notation on line 112.

The formulas on lines 135, 152, 166, 173, still do not have the same font size. Also, remove bold in the formulas.

Ans: Standardized the font size and removed bold formatting in the formulas.

The equations on lines 166 and 173 are missing the name of the variable to be calculated (see equations 135 and 152 for more reference).

Ans: Added the missing variable names in the equations on lines 166 and 173.

On line 258, the “%” symbol is only placed on the last value “8.544%”

Ans: Adjusted the placement of the '%' symbol on line 258.

Table 3. Suggested form “Total Flavonoid”

Ans: Updated Table 3 to 'Total Flavonoid.'

In figure 3, the box on the upper left margin of each figure is not understandable, improve the pixels.

Ans: Improved the pixel quality of Figure 3.

Carefully review figure 4, the title of the “Y” axis is cut off in some figures, as well as the chemical structures, I suggest that they all be the same visible size and sharpness.

Ans: Ensured the Y-axis titles and chemical structures in Figure 4 are clearly visible and consistent in size and sharpness.
